# Exploring microRNAs in Bile Duct Stents as Diagnostic Biomarkers for Biliary Pathologies

**DOI:** 10.3390/cancers17071171

**Published:** 2025-03-31

**Authors:** Noam Mathias Hipler, Cosima Thon, Konrad Lehr, Manuele Furnari, Wilfried Obst, Verena Keitel, Jochen Weigt, Alexander Link

**Affiliations:** 1Department of Gastroenterology, Hepatology and Infectious Diseases, Otto-von-Guericke University Magdeburg, 39120 Magdeburg, Germany; noam.hipler@med.ovgu.de (N.M.H.); cosima.thon@med.ovgu.de (C.T.); konrad.lehr@med.ovgu.de (K.L.); wilfried.obst@med.ovgu.de (W.O.); verena.keitel-anselmino@med.ovgu.de (V.K.); jochen.weigt@med.ovgu.de (J.W.); 2Gastroenterology Unit, Department of Internal Medicine, Policlinico San Martino-IRCCS Hospital, University of Genoa, 16145 Genoa, Italy; manuele.furnari@unige.it

**Keywords:** bile ducts, stents, cholangiocarcinoma, cholestasis, miRNA, cholangitis

## Abstract

The optimal diagnosis of malignant diseases of the bile duct remains a challenge and can affect management and, subsequently, outcomes. Obstructive cholestasis is one of the most common manifestations of both benign and malignant disease and is commonly treated with stent placement. In this proof-of-principle study, we investigated whether specimens from bile duct stents could serve as a source of biomarkers for changes in patients with bile duct disease, including inflammation and malignancy. The data showed that microRNAs can be reliably detected in samples and that the subset of microRNAs may reflect disease states, supporting further screening for the most appropriate targets to diagnose malignant biliary disease.

## 1. Introduction

Diseases of the pancreaticobiliary system include conditions ranging from acute inflammation such as cholangitis and stenosis following procedures such as liver transplantation to severe malignancies such as cholangiocarcinoma (CCA) and pancreatic cancer. In particular, cholangitis is a common complication of cholestasis that requires endoscopic treatment and is associated with poor outcomes [1]. While benign causes of stenosis have an overall relatively good prognosis, CCA is one of the most common conditions requiring appropriate, often personalized management and shows a worse prognosis, partly due to delayed diagnosis [2,3]. Overall, there has been an increase in the incidence of CCA in recent decades, which can be explained by an increase in the prevalence of risk factors such as metabolic syndrome [4]. CCA is usually divided into three main subtypes based on localization, which have different risk factors and incidences [5]. The diagnostic workup for intrahepatic CCA is relatively straightforward and involves identifying the lesion using available imaging modalities followed by histological assessment. However, the diagnosis of extrahepatic CCA can be challenging, despite advances in cytological and histological diagnostics. Obstructive cholestasis usually develops gradually. Obstructive cholestasis is often the first symptom in patients with CCA, which may require diagnostic and therapeutic endoscopic intervention, such as endoscopic retrograde cholangiography (ERC) or cholangioscopy. However, it is often difficult to differentiate between malignant and non-malignant causes of cholestasis, and diagnosis is frequently made at an advanced stage. This delayed diagnosis may also be attributed to the absence of biomarkers and the difficulty in histologically confirming malignant findings, particularly for intraluminal tumors. The frequently used biomarker CA19-9 only has a limited sensitivity and specificity [6] and can also be elevated in other malignant diseases [7] and non-malignant diseases with cholestasis [8].

MicroRNAs (miRNAs) have gained importance as new potential biomarkers for various diseases. MiRNAs with approximately 22 bases are short non-coding RNA sequences that play a role in post-transcriptional regulation and have been associated with a unique stability [9,10,11]. Initially detected in *Caenorhabditis elegans*, the continuous evolution of knowledge has led to an increasing appreciation of miRNAs in terms of cellular function and the development of diseases [12]. Most interestingly, miRNAs are identified in various specimens, and their differential expression has been linked to specific diseases. Furthermore, miRNAs are not only found in mammals or animals, but also in plants, therefore even crossing the biological barrier. In this case, they are called xeno-miRNAs, with potential functional implications [13,14]. Overall, miRNAs can be classified into functional families, such as onco-miRNAs, which are frequently upregulated in malignancies due to their action on proto-oncogenes or tumor suppressor genes that may be downregulated in malignant tissues. For example, miR-21 is frequently linked to neoplastic conditions and has already been detected in bile [15]. Furthermore, increased miR-21 serum levels have been shown in malignant diseases, such as CCA and pancreatic carcinoma [16]. Furthermore, other miRNAs play a crucial role in immunological processes, such as miR-223, which is highly expressed in cells of the myeloid lineage, like polymorphonuclear leukocytes (PMN) [17], and can, therefore, be detected at altered levels in acute inflammatory events and hematological malignancies [18,19]. Taking into account their potential value in the diagnosis of disease, miRNAs have already been explored in bile [15,20,21,22].

Compared to blood and feces, bile has potential advantages for diagnosis due to its close anatomical proximity to specific diseases, which may lead to a better diagnostic performance. However, obtaining bile specimens is not always straightforward, and temporary fluctuations may affect bile miRNA levels. For diagnostic purposes, the highest concentration of exfoliated epithelial cells in bile is of particular interest. Bile duct obstruction often requires the placement of endoscopic stents to alleviate stenosis. It is possible that cells, particularly tumor cells, may adhere to these stents, providing advantages over conventional bile specimens. Since these stents remain in the bile ducts for up to 90 days and are regularly replaced, the adherent content or biofilm may be suitable for analysis.

In this proof-of-principle study, we assessed whether miRNA could be preserved in stents retrieved from patients with bile duct stenosis and potentially serve as biomarkers for differentiating bile duct diseases, particularly stenosis. To this end, we collected bile stents from patients undergoing elective or emergency ERC and conducted analyses of several selected miRNAs.

## 2. Materials and Methods

### 2.1. Study Design

Patients with obstructive bile duct disease undergoing ERC at the Department of Gastroenterology, Hepatology and Infectious Diseases at the Otto-von-Guericke University Magdeburg were asked for agreement that their stent postintervention may be used for research purposes. A total of 212 stents were collected from 112 different patients. Stents from 12 patients were used for preliminary evaluation and optimization purposes or excluded if a minimal set of data was incomplete or missing. The subsequent specimens from 100 patients were included in the final analysis.

Following preanalytical analysis, only one stent per patient was included in the subsequent analysis. The patient cohort consisted of the following groups: malignancies (*n* = 40), 22 of them with CCA, strictures resulting from surgery and liver transplantation, and other diseases such as chronic pancreatitis (*n* = 60) (Table 1). At the time of stent collection, 42 patients had been diagnosed with cholangitis, independent of the underlying disease, based on the Tokyo criteria [23]. Likewise, only patients who had no evidence of cholangitis or cholestasis, independent of the underlying disease, were included in the non-inflammatory comparison group (*n* = 25).

### 2.2. Samples Collection

The stents were collected during ERC and stored at 4 °C for subsequent processing. To prepare the stents, the ends were cut off, and the center and ends were rinsed with 400 µL of nuclease-free water. If clotting with sludge occurred, the inner part of the stent was mobilized using a sterile pipette tip and further eluted in the appropriate nuclease-free water. The eluate was subsequently frozen at −80 °C until further extraction or use.

### 2.3. RNA Isolation and Quantitative Real-Time PCR

Total RNA, including miRNA, was extracted from the stent eluate using the Qiagen miRNeasy mini kit (Qiagen, Hilden, Germany) according to the manufacturers’ instructions, with minor modification [24,25]. Prior to extraction, the stent contents were mixed with QIAzol Lysis Reagent, pipetted into a QIAshredder Spin Column, and centrifuged. The extracted RNA was eluted with 50 µL of nuclease-free water and stored at −80 °C until further analysis. The approximate concentration and purity were determined photometrically. The quantification and measurement of the miRNA expression were carried out using quantitative real-time PCR using TaqMan miRNA assay or SYBR-Green methods. Normalization was performed by using the spike-in method with cel-miR-39. According to the manufacturer’s protocol, approximately 20 ng of total RNA was reverse transcribed, and quantitative real-time PCR analysis was completed using the BioRad CFX Cycler System (Bio-Rad Laboratories, Inc., Hercules, CA, USA). The following primers were used for the analyses: hsa-miR-16-5p (Assay-ID: 000391), cel-miR-39-3p (Assay-ID: 0002000), hsa-miR-223-3p (Assay-ID: 002295) (Thermo Fisher Scientific, Waltham, MA, USA), and hsa-miR-21-5p (miR-21), as previously described [19,26].

### 2.4. Statistical Analysis

The miRNA analyses were performed using threshold cycle (Ct) or 2^−∆Ct^ methods normalized to spiked-in cel-miR-39. Data analysis was performed with GraphPad Prism 10.1 software (GraphPad, San Diego, CA, USA). The D’Agostino test was used to test for normal distribution. The Spearman correlation method was used for correlation analysis. The Mann–Whitney U test was used to test statistical significance in a non-parametric unpaired *t*-test. The Kruskal–Wallis test was used to test statistical significance in an independent non-parametric test for more than two groups. The Friedman test was used to test statistical significance in a dependent non-parametric test for more than two groups.

## 3. Results

Overall, we included 112 patients, although the samples from 12 patients were used for optimization and preanalytical analysis and the final analysis included 100 patients from the representative cohort of patients with malignant or non-malignant disease (Table 1). The selection of miRNAs for the proof-of-principle analysis was based on our previous experience and the targeted selection of potentially tumor-associated miRNAs (miR-21) or inflammation-associated (miR-223) miRNAs and miR-16, commonly used for the normalization of tissue miRNA [27,28]. Specifically, the expression of miR-21 has frequently been reported to be deregulated in CCA [29] and pancreatic cancer [26]. MiR-223, because it is highly expressed in neutrophil granulocytes [18], has been reported to often be deregulated in inflammatory diseases [19].

### 3.1. Preanalytical Evaluation

As this is the first study of miRNA from bile duct stents, our initial objective was to assess the feasibility of measurements and provide a preanalytical evaluation. Our analysis revealed that all miRNAs studied were reproducibly detectable from the bile duct stents. The highest level among the studied miRNAs was observed for miR-223, followed by miR-16, and the lowest quantity was observed for miR-21. All miRNAs were associated with a certain degree of variation (miR-16: 0.01805 ± 0.08992; miR-21: 0.004691 ± 0.02551; and miR-223: 0.03185 ± 0.07980; *p* < 0.0001) (Figure 1A). We also evaluated the following two potential influencing factors: the impact of the stent type and indwelling time on miRNA differences. We analyzed the following three different stent types: double pigtail stents (*n* = 72), Amsterdam stents (*n* = 21), and fully covered metal stents (*n* = 7), as a possible confounding factor. We observed no difference in the miRNA levels between stent types (miR-16: *p* = 0.4272; miR-21: *p* = 0.6650; and miR-223: *p* = 0.2395) (Figure 1B). To evaluate indwelling time, we investigated the correlation between indwelling time and miRNA levels. Our findings indicated that stent indwelling time did not significantly correlate with miRNA levels (miR-16: *p* = 0.4890; miR-21: *p* = 0.2521; and miR-223: *p* = 0.5275) (Figure 1C). Since no normalization marker for stents exists to date, we questioned if miR-16 may potentially be used as a normalizer, as proposed in previous studies [28,30]. The Ct values of miR-16 showed a strong positive correlation with the Ct values of miR-21 (r = 0.8042; 95%-CI: 0.7190 to 0.8656, *p* < 0.0001) and miR-223 (r = 0.7672, 95%-CI: 0.6689 to 0.8392, *p* < 0.0001) (Figure 1D). Considering that miR-16 is also expressed in human and tumor cells and is associated with exfoliated tumor cells, we abandoned the idea to avoid potential bias for the observed association with miR-21 and miR-223. Instead, we optimized our protocol for normalization using spiked-in cel-miR-39, which is commonly applied to blood and fecal specimens [31].

### 3.2. miRNA in Malignant Conditions

In the next step, we investigated whether stents from patients with neoplastic diseases had alterations in their miRNA levels compared to those without neoplastic conditions. Analysis of the studied miRNAs revealed a trend for higher miR-16 and higher miR-223 levels in patients with malignant disease (Figure 2A,B,E,F). It is interesting to note that miR-21 showed a slightly opposite trend, but there were no significant changes between the malignant and non-malignant cohorts (Figure 2C,D). The normalization method did not affect the analysis outcome. We then investigated whether a similar trend could be observed in patients with a proven diagnosis of CCA. Subgroup analysis for the CCA groups showed a similar trend without reaching statistical differences in the miRNAs studied (Figure 2B,D,F).

### 3.3. miRNA in Inflammation

Following the demonstration of the feasibility of miRNA analysis in bile duct stents, we aimed to explore the potential mechanisms that may affect the miRNA levels in stents. Thus, we investigated whether cholangitis, an inflammatory condition of the bile ducts that may lead to an increase in inflammatory cells, affects miRNA levels. To analyze these data, we divided our cohort into the following two groups: those with cholangitis and those without evidence of an inflammatory condition, either clinically or biochemically. Compared to the data related to malignant conditions, specimens from patients with cholangitis showed a strong and consistent difference in miRNA levels compared to the non-inflammatory control group (Figure 3A). Cholangitis was associated with a significant increase in miR-16 (median: 0.002465, 95%-CI: 0.0007770 to 0.007264 vs. 0.000227, 95%-CI: 0.000114 to 0.001270, *p* = 0.0028), increased miR-21 (median: 0.0004610 95%-CI: 0.000145 to 0.00125 vs. 0.0000079 95%-CI: 0.0000274 to 0.000184, *p* = 0.0026), and also increased PMN-associated miR-223 (median: 0.002685 95%-CI: 0.000691 to 0.02673 vs. 0.000667 95%-CI: 0.000196 to 0.00189, *p* = 0.0406) (Figure 3A). Having shown the difference in the miRNA levels in subjects with a confirmed diagnosis of cholangitis, it can be concluded that subsequent changes in the stent environment may be directly related to miRNA changes, further suggesting the need to identify specific miRNAs that may not be dependent on the inflammatory state of bile.

### 3.4. Association of miRNA Levels with Laboratory Data

After demonstrating the potential impact of bile duct disease on miRNA levels, we investigated whether miRNA is associated with laboratory values, including C-reactive protein (CRP), as well as cholestasis markers such as alkaline phosphatase (ALP), gamma-glutamyl transferase (γGT), bilirubin, aspartate aminotransferase (ASAT), and alanine aminotransferase (ALAT) (Figure 3B). The results of the systematic correlation analysis indicated that miR-16 was positively correlated with ALP, γGT, ASAT, ALAT, and bilirubin. Additionally, miR-21 was correlated with γGT, bilirubin, ASAT, and ALAT, while miR-223 only showed a positive association with ALAT, ASAT, and γGT. Notably, no correlations were found for CRP. As mentioned above, this further suggests the impact of cholestasis and potential stent dysfunction on alterations in miRNA levels.

## 4. Discussion

There is an unmet need for novel tools for the diagnosis and treatment of biliary tract diseases, in particular CCA. In this proof-of-principle study, bile duct stents were used as a source of specimens for biomarker analysis. The study successfully demonstrated the detectability of all studied miRNAs (miR-16, miR-21, and miR-223) in the stent specimens. All three miRNAs showed significant differences between patients with cholangitis and those without inflammatory conditions. Furthermore, all three miRNAs showed a positive association with laboratory values, implying the potential value of miRNAs from stents as biomarkers for bile duct disease.

Blood and feces have long been considered as optimal sources for surrogate biomarkers in human disease. Nevertheless, certain diseases are more accurately associated with localized changes, such as molecular alterations in feces for inflammatory bowel disease or colorectal cancer. In this context, there is growing evidence that bile duct diseases, including PSC and CCA, may be best identified through a direct analysis of bile. However, direct bile sampling presents challenges due to limited accessibility and procedural complexity. To address these limitations, we investigated the feasibility of utilizing bile duct stents as an alternative source of biomarkers. This approach not only provides insight into the immediate state of bile, but also captures adherent cells, potentially reflecting long-term alterations within the bile ducts. To facilitate this, we developed an easy-to-implement protocol for collecting samples from bile duct stents, aiming to enhance biomarker discovery and improve diagnostic accuracy for bile-duct-related diseases.

Preanalytical analysis is crucial for evaluating the suitability of biomarkers in specific diseases. In the first part of the study, we investigated the impact of potential factors such as the duration of time since the stent placement and the differences between different stent types on miRNA levels. The results demonstrated that these factors did not influence miRNA expression. The miRNA level was relatively stable and comparable even if we used different stent types, which may potentially be crucial for comparability between different settings and applicability in clinical practice. Additionally, the indwelling time was not considered as a potential confounding factor, as even short-term stent placement revealed reliable miRNA values, suggesting that for diagnostic purposes, even targeted or short-term stent placement, like in settings of PSC, could be a potential solution.

Having shown the detectability of miRNAs from stents, the question arises as to the possible origin of miRNA from stents in the bile duct system. The detectability of miRNA in the bile duct was already reported a decade ago [32]. Some studies were able to identify extracellular vesicles in bile that contained various miRNAs as a potential source of miRNA [20,21]. Thus, this suggests that miRNAs may be among the cells shed from one side, part of the bile, or exosomes. All of these factors, particularly the human and microbial environment, may contribute to stent membrane adhesion, biofilm formation, and may even be partially responsible for stent occlusion. miRNA may, therefore, originate from liver cells or the bile duct epithelium via direct secretion and as part of cells, but probably also from other cells such as inflammatory cells, as we observed a strong pattern of miRNA differences in subjects with cholangitis. Cholangitis, besides bacterial expansion, is also associated with cell damage, including apoptosis, and inflammatory cell infiltration, which may explain the significant increases in all three miRNAs studied in this model. It has been reported in previous studies that, during cholangitis, an increased biofilm production can be observed [33], hypothesizing that biofilm production may be also linked to an increased attachment of biliary molecules, including miRNAs, and, subsequently, higher miRNA expressions. The positive correlations of all three miRNAs with the cholestasis parameters strongly support the suggestion that more miRNAs can become adherent to stents and even be associated with occlusion of the stent. Interestingly, no correlations were found for miR-223, which is commonly found, particularly in immune cells like PMNs and inflammatory markers such as CRP [18].

While the use of bile-derived miRNAs may contribute to the understanding of the pathogenesis of biliary diseases, including PSC and preneoplastic biliary lesions, one of the most interesting aspects of the applicability of this methodology is the miRNAs’ use as potential biomarkers, for example, in neoplastic conditions. Nevertheless, despite the potential differences in miRNA levels in patients with neoplastic conditions, our preliminary data did not reach a statistically significant difference for CCA. In previous studies, many miRNAs (e.g., miR-483-5p and miR-126-3p) were reported in bile, which have been shown to be elevated in CCA [20,22,31,34]. Indeed, these miRNAs may also be detected in bile duct stents. But since this study was intended as an exploratory proof-of-principle project, further studies, for example, using microarrays, will be needed to specifically detect other miRNAs that may be of diagnostic value.

This study focused on the analysis of miRNA levels due to their stability and advantage over other RNA markers. However, there is overall support for the use of stents as a potential biological specimen for other biomarkers. For instance, obtaining tumor DNA from the stent may be another possible approach and potentially used for the identification of malignant disease. For example, the methylation of DNA could be analyzed [35]. The detection of KRAS mutations has already been performed in bile [36], showing promising results. In addition, genes that are characteristic of CCA, such as the IDH and FGFR2 genes [37], could also be investigated as part of a molecular tumor workup. Analyzing the total mutation burden may be an alternative approach for CCA and pancreatic cancer and may be considered in future studies.

This is a proof-of-principle study; therefore, we would like to highlight several limitations that need to be addressed in the future studies. First, the cohort used included subjects from a single center, and comparability in handling and potential influencing factors will be needed for comparison, especially if broader diagnostic applications are to be explored. Another issue is the quality of the RNA, especially when analyzing routine clinical samples. Despite the previously reported stability of miRNA by many groups, the results may still be influenced by preanalytical factors and, therefore, lead to potential variations in RNA quality and quantity. Therefore, further studies may provide valuable insights into the influencing factors. In addition, the lack of an established normalization marker for biliary stents may be an issue. We introduced external spiked-in miRNA, however, this may still not be the perfect solution, as it may not precisely address host-specific differences. Furthermore, it is crucial to perform miRNA profiling studies to identify potential targets that may better reflect the neoplastic cascade in the bile duct system. In fact, a larger sample size, including patients representative of different conditions and probably best characterized for comorbidities and stages, will be essential to increase statistical power, limit potential bias, and ensure robust conclusions. In our setting, all subjects were enrolled regardless of the length of indwelling time, but it is likely that targeted short-term stenting to obtain specimens in non-cholangitis settings would be an alternative option.

## 5. Conclusions

In conclusion, bile duct stents are a promising source for miRNA analysis, providing reliable and reproducible results. Changes in miRNA expressions may be associated with biliary tract diseases, making them potential diagnostic markers once optimized. Further studies using miRNA profiling and validation with independent cohorts are likely necessary to identify the most suitable miRNA biomarker targets for translational applications.

## Figures and Tables

**Figure 1 cancers-17-01171-f001:**
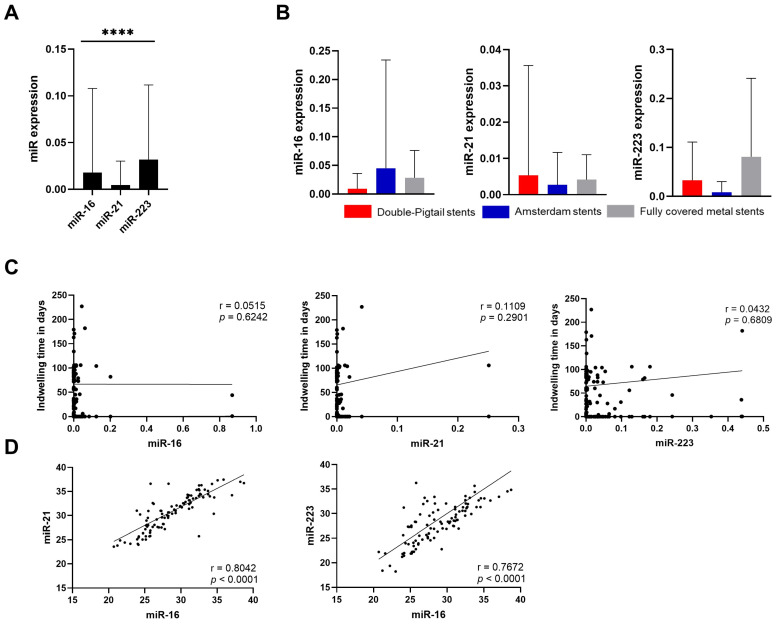
Preanalytical analysis of miRNA in bile duct stents. (**A**) Overall comparison of the expression of the three studied miRNAs from all samples following normalization. Friedman’s test was used for analysis. (**B**) Comparison of the expressions of miRNAs based on the stent type. No significant differences could be shown. Kruskal–Wallis test was used to analyze this. (**C**) Illustration of the correlations between the miRNA expression levels and the indwelling time. No significant correlations could be shown. Analyses were performed using Spearman’s test. (**D**) Correlation of CT values for miR-16 and miR-21, miR-223 from the total cohort. The correlation analyses between miR-16 and miR-21 (r = 0.8042; 95%-CI: 0.7190 to 0.8656, *p* ≤ 0.0001), and miR-223 (r = 0.7672, 95%-CI: 0.6689 to 0.8392, *p* ≤ 0.0001) were performed using Spearman’s test. The miRNA were normalized to spiked-in cel-miR-39 using 2^−∆Ct^-method. The values are shown using mean ± SD. **** *p* < 0.001.

**Figure 2 cancers-17-01171-f002:**
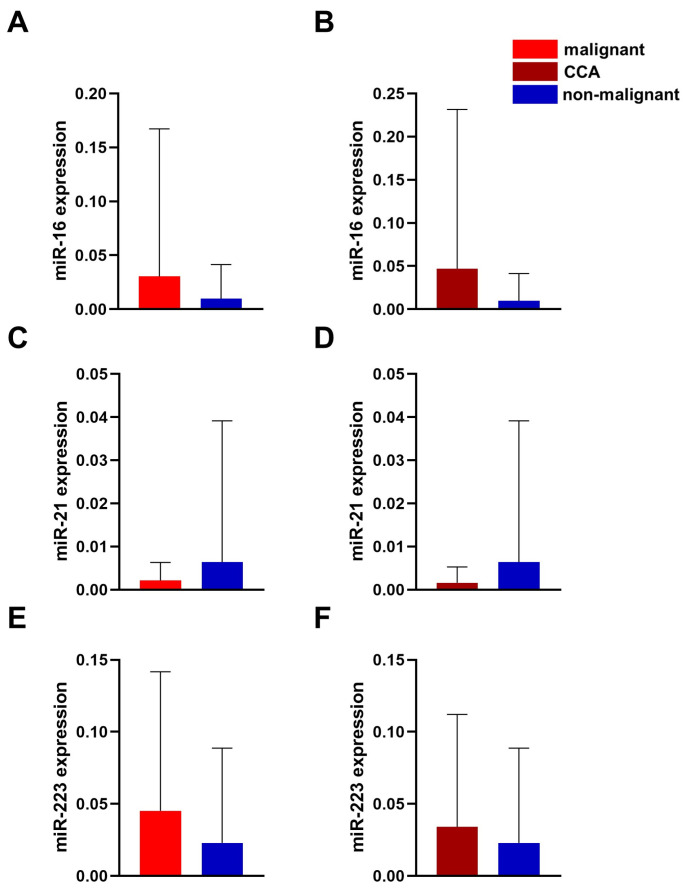
Expression of miR-16 (**A**,**B**), miR-21 (**C**,**D**), and miR-223 (**E**,**F**) in comparison between the malignant cohort, the CCA subcohort and the non-malignant cohort. The miRNAs were normalized for better comparability with cel-miR-39 and 2^∆Ct^-method. The values are shown as mean ± SD. Mann–Whitney test was used to analyze this.

**Figure 3 cancers-17-01171-f003:**
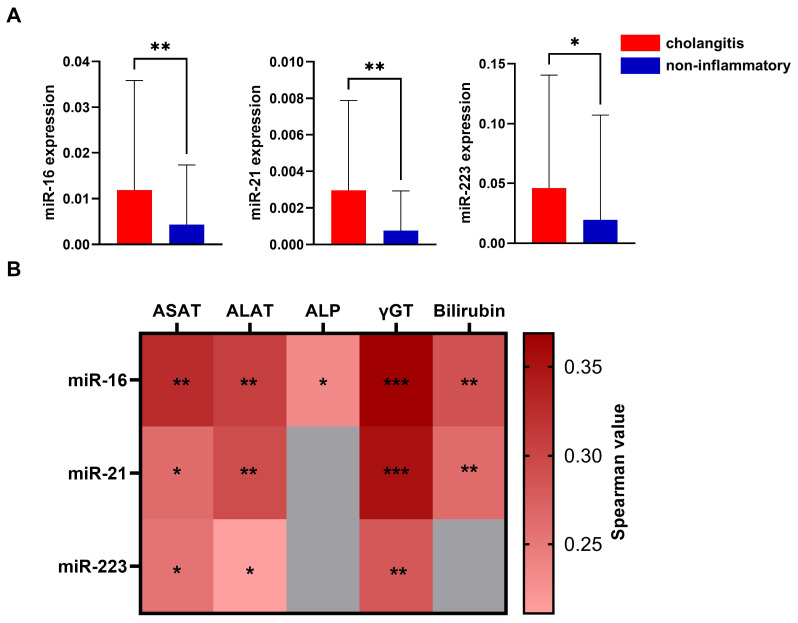
Expression of miR-16, -21, and -223 (**A**) in comparison between the cholangitis cohort and the non-inflammatory cohort. Mann–Whitney test was used for analysis. Expression of miR-16, miR-21, and miR-223 and the correlation with laboratory values (**B**). The miRNAs were normalized for better comparability with cel-miR-39 and 2^−∆Ct^-method. The values are shown using mean ± SD. Correlation analysis were performed using Spearman’s test. *** *p* < 0.001; ** *p* < 0.01; and * *p* < 0.05.

**Table 1 cancers-17-01171-t001:** Characteristics of patients with an indication for stent therapy.

Patients		100
Age in years		65.50 ± 12.57
Sex	Male	71 (71.00%)
	Female	29 (29.00%)
Stent types	Double pigtail	69 (69.00%)
	Amsterdam	21 (21.00%)
	Metal stents	7 (7.00%)
	No information	3 (3.00%)
Indwelling time in days		71.47 ± 63.55
Reason for stent therapy	CCA	22
	PCA	5
	CRC	5
	Other malignancies	8
	Non-malignant stenosis	60
Patients with cholangitis		42
Control group		25

Abbreviations: CCA: cholangiocarcinoma, PCA: pancreatic cancer, CRC: colorectal cancer; metal stents include fully covered or uncovered self-expanding metal stents.

## Data Availability

The study data can be obtained from the corresponding author on reasonable request.

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
