# Peer review of "Exploring microRNAs in Bile Duct Stents as Diagnostic Biomarkers for Biliary Pathologies"

_cancers, 2025, doi:10.3390/cancers17071171_

Round 1
Reviewer 1 Report
Comments and Suggestions for Authors
The overall level of the paper is good. The topic is interesting, the aim is clear and I personally appreciate the effort of the authors in setting up a new methodology to investigate miRNAs in bile duct stents, considering the lack of validated endogenous controls and of previous results. However, the following major and minor issues should be addressed before publication.
Major issues:
- Please provide a better explanation on how the samples were stratified in the different groups. This information is reported, but it is difficult to follow.
- It is not clear by which method the authors analyzed the qPCR data. They refer to “2deltaCt-method” (line 259) or 2ΔCt methods (line 159). Do you mean 2^-delta ct method? This is the correct method to apply when analyzing Ct data. I am concerned about the results because the lack of a “minus” in the formula changes the interpretation of the results.
- In figures 2 A-B-D and 3 (A to F) what does the y axis refer to? 2^-delta ct level of the miRNAs?
- Please, provide a better explanation of the figure 2A? I can not understand the meaning of this analysis.
- In the legend of each figure, where a p-value is indicated, the author should mention the statistical test applied.
- “miRNAs concentration levels” should be replaced with “miRNAs expression levels”. You are not testing the concentration.
- In the text are reported the median expression values of the miRNAs, while in the figure other values are reported. Do the graph bars represent the mean and standard deviation of miRNA expression? If so, please clarify this issue in the legend.
Minor issues:
- In the legend of Figure 2 and 3 the extended name of miRNAs should be indicated (for example “miR-16 and miR-21, -223” at line 208 should be replaced with “miR-16 and miR-21, miR-223”
The English could be improved
Author Response
Comments and Suggestions for Authors
The overall level of the paper is good. The topic is interesting, the aim is clear and I personally appreciate the effort of the authors in setting up a new methodology to investigate miRNAs in bile duct stents, considering the lack of validated endogenous controls and of previous results. However, the following major and minor issues should be addressed before publication.
Major issues:
Please provide a better explanation on how the samples were stratified in the different groups. This information is reported, but it is difficult to follow.
Response: We thank you for this important point and apologize for the missing information. Overall, the cohort was divided into a malignant and non-malignant group based on underlying disease independent of cholangitis or stent dysfunction. For a second analysis, patients with cholangitis were classified based on the Tokyo guidelines. A non-inflammatory comparator cohort was defined as: no laboratory cholestasis and or elevated inflammatory values. Patients who had onset of stent dysfunction, cholestasis or inflammation were not included in this analysis; therefore the comparison is 25 vs 40. We have added this information to the text (line 116-120).
It is not clear by which method the authors analyzed the qPCR data. They refer to “2deltaCt-method” (line 259) or 2ΔCt methods (line 159). Do you mean 2^-delta ct method? This is the correct method to apply when analyzing Ct data. I am concerned about the results because the lack of a “minus” in the formula changes the interpretation of the results. In figures 2 A-B-D and 3 (A to F) what does the y axis refer to? 2^-delta ct level of the miRNAs?
Response: We used the 2^-delta method for the analysis. We have now clarified this in the material and methods section and changed the labeling and description of the graphs accordingly.
Please, provide a better explanation of the figure 2A? I can not understand the meaning of this analysis.
Response: We would like to clarify the message. In order to provide an overall view of the expression of miRNA in the bile duct samples, we have included this summary of miRNA expression in the bile duct samples. While very high levels of miR-21 and miR-16 have been reported for certain samples such as blood or faeces, this does not seem to be the case for the bile samples where the high level of miR-223 is present. In addition, the other message is to provide readers with the overall detectability of miRNA in the study samples, as this is a first part of the proof-of-principle study. We hope to have clarified the message in Figure 2a (now 1a).
In the legend of each figure, where a p-value is indicated, the author should mention the statistical test applied.
Response: Thanks for this valuable comment. We added this information to the script.
“miRNAs concentration levels” should be replaced with “miRNAs expression levels”. You are not testing the concentration.
Response: We appreciate your comments on this issue. It is thought that miRNA expression may be better suited to the tissues and sources where active miRNA expression occurs. Increasingly, many groups are referring to the sources where there is no active miRNA expression, such as bile, faeces or, in our case, the sediment in the stent, with concentration being a preferred term. However, we are happy with both terms and have changed concentration to expression as requested.
In the text are reported the median expression values of the miRNAs, while in the figure other values are reported. Do the graph bars represent the mean and standard deviation of miRNA expression? If so, please clarify this issue in the legend.
Response: We apologise for the misleading reporting and the explanation is included in the revised version. The graphs show the mean +/- SD. This is due to the large range of scatter. Therefore, the mean has been used for clarity. Missing information has been added to the figure legends.
Minor issues:
In the legend of Figure 2 and 3 the extended name of miRNAs should be indicated (for example “miR-16 and miR-21, -223” at line 208 should be replaced with “miR-16 and miR-21, miR-223”
Response: Thank you. We corrected this and used the extended name in the legends of figure 1 and 2.
Reviewer 2 Report
Comments and Suggestions for Authors
In this study, the authors attempted to provide proof-of-concept that microRNAs (miRNAs) detected from bile duct stents could be biomarkers of biliary pathologies including inflammation and malignancy. They provide evidence that multiple miRNAs were successfully detected from the bile duct stent samples, and show that miRNA levels are variable among disease states. This study is well designed, and results are clear. I suggest several minor issues which should be addressed before final acceptance.
Major point
As the authors stated, expression levels of the miRNAs (miR-16, miR21, and miR223) are quite variable among samples. Thus, although there are some tendencies in the miRNA expression differences among disease states, they are not statistically significant. When analyzing RNA samples extracted from clinical specimens, quantities and qualities of the RNA could significantly affect the results of gene expression analysis. Authors should validate whether the miRNA expression variabilities reflect the variations in the quantities and qualities (e.g. RNA integrity number, RIN) of the RNA samples. Alternatively, the authors may discuss the possibility that quantities and qualities of the RNA samples may have affected the results in the current study.
Minor points
Page 2, line 70: “Non-coding microRNAs” may be simply corrected to “MicroRNAs”, because it is widely known that miRNAs are non-coding RNAs.
Page 5, line 183: The sentence “Highest was level among studied miRNA was observed for miR-223 followed by miR-16 and lowest quantity for miR-21” is grammatically incorrect. It may be corrected to “The highest level among the studied miRNAs was observed for miR-223, followed by miR-16, and the lowest quantity was observed for miR-21”.
Page 5, lines 192 and 193: “microRNA” may be corrected to “miRNA”.
Author Response
Comments and Suggestions for Authors
In this study, the authors attempted to provide proof-of-concept that microRNAs (miRNAs) detected from bile duct stents could be biomarkers of biliary pathologies including inflammation and malignancy. They provide evidence that multiple miRNAs were successfully detected from the bile duct stent samples, and show that miRNA levels are variable among disease states. This study is well designed, and results are clear. I suggest several minor issues which should be addressed before final acceptance.
Reponse: We kindly appreciate the positive and constructive feedback.
Major point
As the authors stated, expression levels of the miRNAs (miR-16, miR21, and miR223) are quite variable among samples. Thus, although there are some tendencies in the miRNA expression differences among disease states, they are not statistically significant. When analyzing RNA samples extracted from clinical specimens, quantities and qualities of the RNA could significantly affect the results of gene expression analysis. Authors should validate whether the miRNA expression variabilities reflect the variations in the quantities and qualities (e.g. RNA integrity number, RIN) of the RNA samples. Alternatively, the authors may discuss the possibility that quantities and qualities of the RNA samples may have affected the results in the current study
Response: Thank you for pointing this out. According to the request, we have added the following section as part of the discussion: “Another issue is the quality of the RNA, especially when analyzing routine clinical samples. Despite the previously reported stability of miRNA by many groups, the result may still be influenced by pre-analytical factors and therefore lead to potential variations in RNA quality and quantity. Therefore, further studies may provide valuable insights into the influencing factors.” (Line 378-382).
Minor points
Page 2, line 70: “Non-coding microRNAs” may be simply corrected to “MicroRNAs”, because it is widely known that miRNAs are non-coding RNAs
Response: Thank you, the point has corrected.
Page 5, line 183: The sentence “Highest was level among studied miRNA was observed for miR-223 followed by miR-16 and lowest quantity for miR-21” is grammatically incorrect. It may be corrected to “The highest level among the studied miRNAs was observed for miR-223, followed by miR-16, and the lowest quantity was observed for miR-21”.
Response: Thank you for your comment. We corrected the sentence.
Page 5, lines 192 and 193: “microRNA” may be corrected to “miRNA”.
Response: Thank you, we changed microRNA to miRNA.
Round 2
Reviewer 1 Report
Comments and Suggestions for Authors
The authors made all the suggested corrections. I agree with the publication of the manuscript in the present form.